# CoMIR: Contrastive Multimodal Image Representation for Registration

**Nicolas Pielawski**[*], **Elisabeth Wetzer**[*], **Johan Öfverstedt,**
**Jiahao Lu, Carolina Wählby, Joakim Lindblad, Nataša Sladoje**

Dept. of Information Technology, Uppsala University, Sweden

## Abstract

We propose contrastive coding to learn shared, dense image representations, referred to as CoMIRs (Contrastive Multimodal Image Representations). CoMIRs enable the registration of multimodal images where existing registration methods often fail due to a lack of sufficiently similar image structures. CoMIRs reduce the multimodal registration problem to a monomodal one, in which general intensity-based, as well as feature-based, registration algorithms can be applied. The method involves training one neural network per modality on aligned images, using a contrastive loss based on noise-contrastive estimation (InfoNCE). Unlike other contrastive coding methods, used for, e.g., classification, our approach generates image-like representations that contain the information shared between modalities. We introduce a novel, hyperparameter-free modification to InfoNCE, to enforce rotational equivariance of the learnt representations, a property essential to the registration task. We assess the extent of achieved rotational equivariance and the stability of the representations with respect to weight initialization, training set, and hyperparameter settings, on a remote sensing dataset of RGB and near-infrared images. We evaluate the learnt representations through registration of a biomedical dataset of bright-field and second-harmonic generation microscopy images; two modalities with very little apparent correlation. The proposed approach based on CoMIRs significantly outperforms registration of representations created by GAN-based image-to-image translation, as well as a state-of-the-art, application-specific method which takes additional knowledge about the data into account. Code is available at: `https://github.com/MIDA-group/CoMIR`.

---

[*] Authors contributed equally.

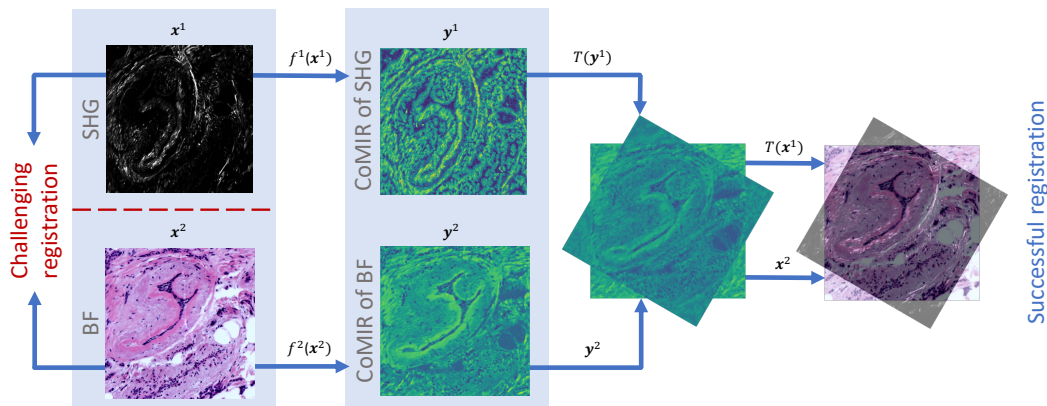

Figure 1: Registration of images of different modalities (here bright-field (BF) and second-harmonic generation imaging (SHG)) may be very challenging. CoMIR successfully estimates the shared representation of these images, and enables their successful registration by monomodal approaches.

# 1 Introduction

Multimodal images refer to images captured with multiple types of sensors, where each sensor outputs information not fully provided by the other sensors. Multimodal image fusion is the process of combining information from multiple imaging modalities. It allows downstream tasks to exploit complementary information as well as relationships between modalities. In order to perform image fusion, the images need to be aligned, either by joint acquisition, or by manual or automated registration. To enable registration, common structures between the modalities need to be found, reflected by the shared or mutual information (MI); this is in general a very difficult task. Our method, *Contrastive Multimodal Image Representation for registration (CoMIR)* reduces the challenging problem of multimodal registration to a simpler, monomodal one, as shown in Fig. 1. While image-to-image translation aims to predict cross-modality, i.e. to predict one modality given the other modality as input, our method directly learns representations called CoMIRs, relying on the MI by maximizing the Mutual Information Noise-Contrastive Estimation (InfoNCE) [20, 25]. InfoNCE can, under certain assumptions, act as a lower bound to MI, and has been successfully applied in connection with contrastive coding in other tasks, such as classification and segmentation. Although contrastive losses (CL) are often used for representation learning [1, 6, 22, 24, 25, 43, 46, 47, 49, 53, 62], to the best of our knowledge, the method presented here is the first to produce dense representations for very different imaging modalities which can be utilized for monomodal registration by existing feature- or intensity-based registration methods. InfoNCE has been previously used to learn embeddings used in classfication and segmentation tasks in which the resulting subspace is required to feature properties such as separability between classes. Furthermore, registration requires representations which are both translation and rotation equivariant, as opposed to classification, for which invariance is required. Our proposed method produces image-like, contrastive representations that possess the necessary equivariant properties to find a transformation between the original inputs.

The contributions of this paper are the following: We show that (1) contrastive learning of aligned pairs of images can produce representations that reduce complex (or even non-feasible), rigid multimodal registration tasks to much simpler monomodal registration; (2) the proposed CoMIRs are rotation equivariant due to a modification of a commonly used CL. This modification is model-independent and does not require any architecture modifications, nor any additional hyperparameter tuning. Although the method is supervised, thanks to our sophisticated scheme for generating training patches, as little as one image pair can be sufficient to generate CoMIRs (depending on the nature of the imaging data).

# 2 Background and Related Work

Image registration consists of finding transformations matching unaligned images. This process is of highest importance, notably in computer vision and medical imaging, and has been extensively studied in the last fifty years [19]. In many cases, a modality-independent solution can be obtained by optimizing MI or generating hand-crafted local structural feature descriptors [23]. Unfortunately, optimizing MI has many caveats: it has a slow convergence rate and a narrow catchment basin with many local maxima. Furthermore, the method relies on specific statistical clues that not all datasets contain. Learning-based algorithms can potentially provide a solution, as the algorithm can learn semantic relationships between modalities. In general it is difficult to find suitable image similarities between multimodal images due to their difference in appearance and/or signal expression. In [58] internal similarities in images across modalities are captured by defining structural representations of image patches through Laplacian eigenmaps and entropy images in order to perform multimodal registration. [5] uses sparse coding to learn a mapping between two image modalities such that sum of squared differences or MI can be used as measures for image registration. Deep learning has changed the field of similarity and metric learning from learning distances to learning feature embeddings that fit distance functions [18]. CL [21] and in particular triplet loss [48, 61] have shown great success in deep metric learning for a wide range of applications, such as object retrieval [45], single-shot object classification [55], object tracking [34, 52], classification [8] and multimodal patch matching of aerial images [63]. In [26], keypoint descriptors are learnt using contrastive metric learning to minimize the difference between two feature representations from two corresponding points and maximize the difference of two feature representation from two distant points to register multimodal medical images. The link between CL, classification and MI is an active area of study, with [49] showing that the triplet loss can be extended to make use of $N$ negative samples instead of one and is an approximation of the softmax function.

**Contrastive Loss and MI** Several recent representation learning approaches [1, 24, 25, 46, 51, 53, 62] are based on the infomax principle which refers to maximizing the MI between input and output, [2, 39]. One approach showing impressive results is Deep InfoMax (DIM) [25] which argues that maximizing MI with high dimensional inputs and outputs is a difficult task. Similarly, Contrastive Predictive Coding (CPC)[46] maximizes MI between global and local representation pairs by sequential aggregation to learn latent features which can be used for classification. Using Noise-Contrastive Estimation (InfoNCE) as a lower bound to MI with flexible critic was found to result in the most useful representations for classification in [25, 46]. It was further used in [1], where InfoNCE is maximized between features from multiple views in a self-supervised manner. The term multiview can refer to augmented versions of one unlabelled image or multiple modalities of one instance. Instead of maximizing MI between features from one single image as in DIM, MI is maximized across multiple feature scales simultaneously as well as across independently augmented copies of each image. However, in [54], Tschannen et al. show that maximizing tighter bounds on MI than InfoNCE can result in worse representations and argue that the success of these methods cannot only be attributed to the maximization of MI. The authors put the success of approximate MI maximization in connection with the usage of the triplet loss or CL by showing that representation learning across different views by maximizing InfoNCE can be equivalent to metric learning using the multi-class K-pair loss presented in [49]. Tschannen et al. point out that the negative samples for the CL have to be drawn independently in order for InfoNCE to yield a lower bound for the MI, an assumption often disregarded. Despite this, [1, 6, 25, 47, 53] report a better performance of the respective downstream task when using many negative samples or hard examples.

**Equivariance** As [25] emphasizes, the usefulness and quality of a learnt representation is not only a matter of information content but also representational characteristics. Features desirable to request from a representation are equivariances. In tasks such as segmentation or registration, translational and rotational equivariance are highly beneficial. Equivariance defines the property of a function to commute with the action of a symmetry group when its domain and codomain are acted on by that symmetry group. A function or operator $f : \Omega \mapsto Y$ is called equivariant under a family of transformations $\mathcal{T}$ if for any transformation $T \in \mathcal{T}$, there exists $T' \in \mathcal{T}$ s.t.

$$f(T(X)) = T'(f(X)) \quad \forall X \in \Omega. \tag{1}$$

Feature equivariance can be achieved by three different approaches. Firstly, by data augmentation, where randomly transformed pairs of input and label masks are passed to a model which learns some degree of equivariance. The second approach is model-based, as proposed in [12, 13, 60], where equivariant mappings are achieved by adjusting the convolutional, activation and pooling layers to be applied over groups and sharing weights. Thereby these so-called group equivariant convolutional networks (G-CNNs) ensure that the layers themselves become equivariant. A further in-depth study of the theory of equivariant CNNs is presented in [11] and [59]. Thirdly, [9] encourages rotational invariance through an additional constraint in the loss function which has been adapted by [37] to achieve equivariance. They propose an additional term to the cross-entropy loss used to train the model for segmentation: $L_{rot} = \frac{1}{2N} \sum_{x_i \in X} ||O(I) - \bar{r}(I)||_2^2$, where $O(I)$ is the feature map of the image at $0°$ and $\bar{r}(I)$ the mean feature map of the input rotated by multiples of $90°$. This introduces a hyperparameter, impairs training due to the different scale of gradients between the cross-entropy loss and $L_{rot}$ and cannot guarantee any rotational equivariance apart from the $\mathcal{C}_4$ symmetry group, i.e. rotations by multiples of $90°$. Enforcing equivariance through the loss was used in [33], which modifies a CL proposed in [56] to learn action-equivariance for Markov Decision Processes.

## 3 Proposed Method

We introduce a modality independent approach to map two given images of different modalities to similar representations called CoMIRs. Contrastive learning is applied to aligned pairs of images during training to create dense representations. These learnt representations are sufficiently similar to allow the application of monomodal registration algorithms. As our method does not require any additional knowledge regarding the modalities at hand, there are no limitations with respect to application area. In the following section we introduce the CL used in our method, the sampling scheme needed to form the CL, and our modifications to achieve rotational equivariance. We also discuss the choice of critic used for the downstream task of registration.

**Contrastive Loss** Here, we introduce the CL for two modalities, the general case for $M$ modalities is given in App. 7.1. Let $\mathcal{D} = \{(\boldsymbol{x}_i^1, \boldsymbol{x}_i^2)\}_{i=1}^n$ be an i.i.d. dataset containing $n$ data points, where $\boldsymbol{x}^j$ is an image in modality $j$, and $f_{\boldsymbol{\theta}_j}$ the network processing modality $j$ with respective parameters $\boldsymbol{\theta}_j$

for $j \in \{1, 2\}$. For an arbitrary datapoint $\boldsymbol{x} = (\boldsymbol{x}^1, \boldsymbol{x}^2) \in \mathcal{D}$, the loss is given by

$$\mathcal{L}^{opt}(\mathcal{D}) = -\mathbb{E}_{(\boldsymbol{x}^1,\boldsymbol{x}^2)\sim\mathcal{D}} \left[ \log \frac{\frac{p(\boldsymbol{x}^1,\boldsymbol{x}^2)}{p(\boldsymbol{x}^1)p(\boldsymbol{x}^2)}}{\frac{p(\boldsymbol{x}^1,\boldsymbol{x}^2)}{p(\boldsymbol{x}^1)p(\boldsymbol{x}^2)} + \sum_{\boldsymbol{x}_i \in \mathcal{D}\setminus\{\boldsymbol{x}\}} \frac{p(\boldsymbol{x}_i^1,\boldsymbol{x}_i^2)}{p(\boldsymbol{x}_i^1)p(\boldsymbol{x}_i^2)}} \right] \tag{2}$$

Eq. (2) is equivalent to a categorical loss discriminating between negative and positive samples. The ratio distribution $\frac{p(\boldsymbol{x}^1,\boldsymbol{x}^2)}{p(\boldsymbol{x}^1)p(\boldsymbol{x}^2)}$ is approximated by the exponential of a critic function $h(\boldsymbol{y}^1, \boldsymbol{y}^2)$ that computes (some arbitrary) similarity between CoMIRs $\boldsymbol{y}^1 = f_{\boldsymbol{\theta}_1}(\boldsymbol{x}^1)$ and $\boldsymbol{y}^2 = f_{\boldsymbol{\theta}_2}(\boldsymbol{x}^2)$ for the scaling parameter $\tau > 0$

$$\mathcal{L}_{\boldsymbol{\theta}}(\mathcal{D}) = -\frac{1}{n} \sum_{i=1}^{n} \left( \log \frac{e^{h(\boldsymbol{y}_i^1,\boldsymbol{y}_i^2)/\tau}}{e^{h(\boldsymbol{y}_i^1,\boldsymbol{y}_i^2)/\tau} + \sum_{\boldsymbol{y}_j^1,\boldsymbol{y}_j^2 \in \mathcal{D}_{neg}} e^{h(\boldsymbol{y}_j^1,\boldsymbol{y}_j^2)/\tau}} \right) \tag{3}$$

$\mathcal{L}(\mathcal{D})$ is named InfoNCE as described in [46]. Its minimization approximately maximizes a lower bound on the MI, given by $I(\boldsymbol{y}^1, \boldsymbol{y}^2) - \log(n)$, which gets tighter as $n \to \infty$ [46, App. A1], assuming that $\mathcal{D}_{neg}$, the set of chosen negative samples, is sampled i.i.d. The MI is defined as $I(\boldsymbol{x}, \boldsymbol{y}) = \mathbb{E}_{\boldsymbol{x}\sim\mathcal{X}, \boldsymbol{y}\sim\mathcal{Y}}[\log \frac{p(\boldsymbol{x},\boldsymbol{y})}{p(\boldsymbol{x})p(\boldsymbol{y})}]$.

**The Critic** The loss function in Eq. (2) contains the ratio $\frac{p(\boldsymbol{x}^1,\boldsymbol{x}^2)}{p(\boldsymbol{x}^1)p(\boldsymbol{x}^2)}$ which is an unknown quantity [62], but can be approximated with the exponential of a statistical, often bilinear, model [1, 6, 46, 49, 53, 62]. Our experiments using a bilinear model resulted in CoMIRs less suitable for registration (see section 4.2 and App. 7.4, Fig. 11) than choosing a positive, symmetric critic $h(\boldsymbol{y}^1, \boldsymbol{y}^2)$ with a global maximum for $\boldsymbol{y}^1 = \boldsymbol{y}^2$. We experiment with both a Gaussian model with a constant variance $h(\boldsymbol{y}^1, \boldsymbol{y}^2) = -||\boldsymbol{y}^1 - \boldsymbol{y}^2||_2^2$ which uses the mean squared error (MSE) as a similarity function, and a trigonometric model $h(\boldsymbol{y}^1, \boldsymbol{y}^2) = \frac{\langle \boldsymbol{y}^1, \boldsymbol{y}^2 \rangle}{||\boldsymbol{y}^1|| \, ||\boldsymbol{y}^2||}$ which relates to the cosine similarity.

**Rotational Equivariance** To equip our model with rotational equivariance, we propose to add to our objective function an additional constraint which does not require additional parameter tuning and can be incorporated into the CL. In particular, instead of only maximizing $h(\boldsymbol{y}_i^1, \boldsymbol{y}_i^2)$ within the CL for an aligned pair $\boldsymbol{x}_i^1, \boldsymbol{x}_i^2$, we also maximize $h(\boldsymbol{y}_i^1, T_1'(f_{\boldsymbol{\theta}_1}(T_1(\boldsymbol{x}_i^1))))$, and $h(\boldsymbol{y}_i^2, T_2'(f_{\boldsymbol{\theta}_2}(T_2(\boldsymbol{x}_i^2))))$ for $f_{\boldsymbol{\theta}_i}$ being the model trained on modality $i$, $h$ the critic between the resulting representations, and $T_i, T_i' \in \mathcal{T}$. Here we choose $\mathcal{T} = \mathcal{C}_4$, the finite, cyclic, symmetry group of multiples of $90°$ rotations. While the actions of this symmetry group do not require any interpolation of the input images or CoMIRs, they result in sufficient rotational equivariance for angles beyond multiple of $90°$, as shown in section 4.1. In general, any symmetry group could be chosen. Rather than extending our original loss term $h(\boldsymbol{y}_i^1, \boldsymbol{y}_i^2)$ by three separate explicit loss terms, we combine the constraints in a single loss term that implicitly enforces the $\mathcal{C}_4$ equivariance

$$h(T_1'(f_{\boldsymbol{\theta}_1}(T_1(\boldsymbol{x}_i^1))), T_2'(f_{\boldsymbol{\theta}_2}(T_2(\boldsymbol{x}_i^2)))). \tag{4}$$

We randomly sample $T_1$ and $T_2$ once per training step for each element in the batch, which iteratively optimizes all combinations of $\mathcal{C}_4$-transformations.

**Sampling of negative samples** In [32], the authors argued that the ability to discriminate between signal and noise increases with more negative samples. We sample negative samples $\mathcal{D}_{neg}$ from random patches during training within the entirety of all training image pairs at random positions with random orientations. The patches can be extracted from the original images without introducing any padding border effects caused by rotations. Every extracted patch is subject to data specific, random augmentation. This sampling scheme guarantees large variation within every batch, which increases the (statistical) efficiency and generalization of the model. The elements of both modalities from the batch are reused as negatives, such that for a given pair of matching samples, there are $2n - 2$ negative samples available, for a batch size of $n$ pairs, as done in [6].

## 4 Experiments

We evaluate our proposed method on two multimodal image datasets.

**Zurich Dataset** The open Zurich dataset [57] consists of 20 aerial images of the city of Zurich of about $930 \times 940$px. The images are composed of four channels, RGB and Near-InfraRed (NIR) and are captured with the same sensor in identical resolution. An example is given in App. 7.2, Fig. 5.

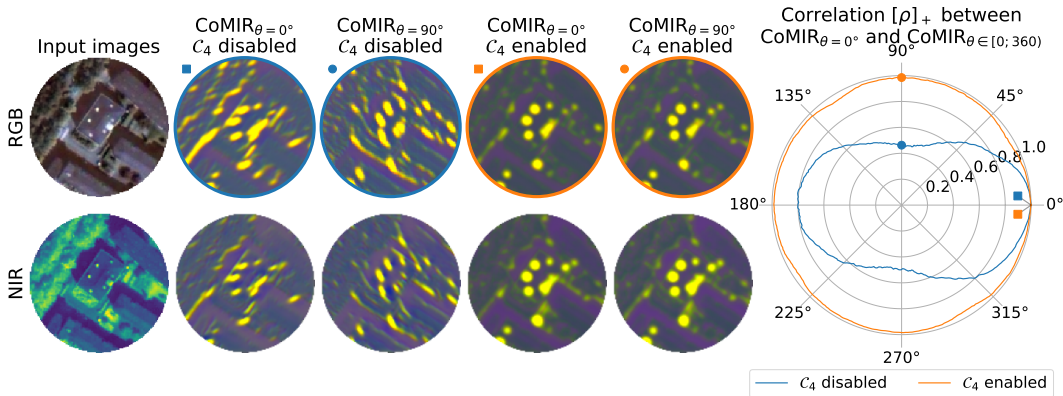

Figure 2: Top row: RGB input patch $x^1$ cropped from the Zurich test set and the resulting CoMIRs using the cosine similarity; Bottom row: the matching NIR image $x^2$ and its resulting CoMIRs. The stabilized CoMIRs $f_{\theta_i}(x^i)$ and $f_{\theta_i}(T(x^i))$ are shown for $x^i$, $T$ a rotation of 90°. Without the $\mathcal{C}_4$-equivariance constraint in the loss function, the resulting CoMIRs (marked blue) are not identical and display edge artifacts. The polar plot (to the right) shows the positive correlation between the stabilized $\text{CoMIR}_{\theta=0°} = f_{\theta_i}(x^i)$ and $\text{CoMIR}_\theta = f_{\theta_i}(T(x^i))$ for $T$ a rotation by $\theta \in [0, 360)°$, demonstrating that rotational equivariance beyond multiples of 90° is achieved by the $\mathcal{C}_4$-constrained CL. Animated version: `https://youtu.be/iN5GlPWFZ_Q`.

**Biomedical Dataset** The dataset consists of 206 aligned Bright-Field (BF) and Second-Harmonic Generation (SHG) tissue microarray core image pairs which are $2048 \times 2048$px in spatial dimensions and were manually aligned using landmarks [30]. The training set consists of 40 image pairs of size $834 \times 834$px which are center-cropped from the corresponding original images. The validation set for the CNN training consists of another 25 such pairs, a tuning set for registration parameters of additional 7 pairs, and the test set for evaluation of another 134 pairs. Fig. 6 in App. 7.2 shows an example image of an original tissue microarray core and the center-cropped patch used in this study. The cropped dataset used in this study is provided in [16].

## 4.1 Evaluation of Representations

**Rotational Equivariance** While different downstream tasks might require different properties of the learnt representations, for registration, we require the representations to be rotationally equivariant. The impact of the rotational equivariance constraint on the learnt CoMIRs of the Zurich dataset is displayed in Fig. 2. When the rotation equivariance is not explicitly enforced in the model or in the loss, the CoMIRs feature edge artifacts that rotate along with the input. We test the level of equivariance by measuring the positive correlation between the stabilized CoMIR of an input image and the stabilized CoMIR of the rotated input image. The result for $\theta \in [0, 360)°$ is shown in Fig. 2 and demonstrates that the $\mathcal{C}_4$-constrained CL achieves rotational equivariance for all angles, not being restricted to only multiples of 90°.

**Reproducibility of CoMIR** Training a model to produce CoMIRs with $c$ channels leads to multiple solutions reaching the same loss value. One such case is a permutation of the channels, and is likely to occur between two training sessions; $c!$ different permutations can be achieved. It is possible, however, to consistently obtain very similar CoMIRs between experiments. [10] observe that the convergence of convolutional filters is in direct connection with their initialization. Similarly, we find that initializing the models' parameters randomly with a fixed seed results in similarly trained models and CoMIRs. We compare CoMIRs generated by 50 identical models, all initialized with the same seed and trained on the Zurich dataset. We then compute the mean pairwise correlation between all the CoMIRs to measure the similarity and consistency between runs. The source of stochasticity between runs originates from the minibatch sampling and from the data augmentation scheme. To test the influence of the data we train 20 models on patches from the same single image, and additionally train 19 models on patches augmented from a single training image varying for each experiment. Table 1 shows the results of the experiments. We observe a high consistency between training runs when the training patches are sampled from the same image and the model's parameters are initialized with the same seed. Fig. 10 in App. 7.4 shows the generated CoMIRs used to compute the average correlation.

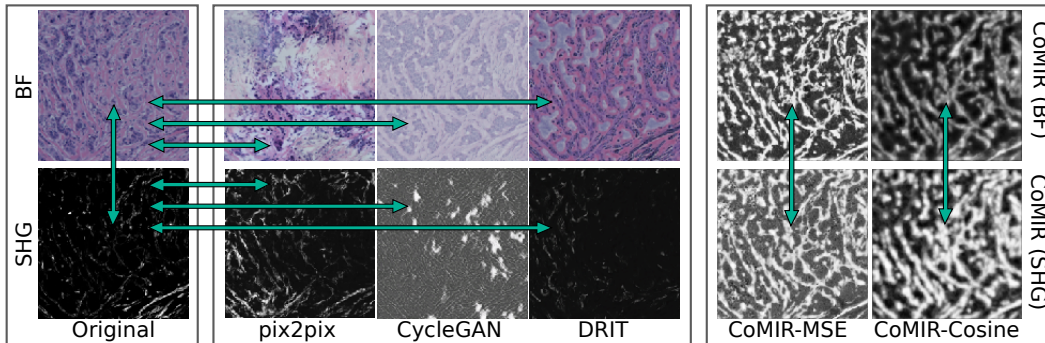

Figure 3: Multiple methods to transform BF and SHG into one common modality. To the left, the original BF (top) and SHG (bottom) images from the test set, followed by image translations of BF and SHG into the respectively other domain. To the right CoMIRs based on MSE and cosine similarity. The arrows indicate the different image pairs to be registered in the experimental setup.

Table 1: Mean pairwise correlation between the CoMIRs during 4 different experiments on the Zurich dataset. The consistency (i.e. the reproducibility of a given training session) between the CoMIRs is highest when the weights are initialized with a fixed seed (most important) and the training distribution is similar (less important). 95% C.I. estimated with empirical bootstrap.

| Mean pairwise Corr. | 1 unique train image | 1 varying train image per model |
|---|---|---|
| Fixed seed init. | 93.0% [ 90.6%; 95.5%], N=100 | 78.4% [ 70.4%; 83.9%], N=38 |
| Random seed init. | -2.7% [-22.3%; -1.4%], N=40 | 1.9% [-18.0%; 5.7%], N=38 |

**Temperature Analysis** The temperature $\tau$ is a hyperparameter of the loss function. High $\tau$ yields a smoother loss. We tested a wide range of values of $\tau$ and observed that all models converged and produced reasonable CoMIRs. See App. 7.4, Fig. 11, for examples of representations with various $\tau$ and loss combinations. We found $\tau = 0.1$ to be a good setting for the Zurich and $\tau = 0.5$ for the biomedical dataset.

## 4.2 Multimodal Image Registration

**Evaluation Set and Metrics for Registration** For each image pair in the test set of 134 biomedical images, first a random rotation up to $\pm 30$ degrees, followed by random translations in x and y directions of up to $\pm 100$px was applied. To avoid any border effects, the transformed patches were cropped from the original $2048 \times 2048$ images. The magnitude of transformation was measured by the average Euclidean displacement $\frac{1}{4} \sum_{i=1}^{4} ||C_i^{Ref} - C_i^{Trans}||_2$ of the corner points $C_i$ between the reference image $C_i^{Ref}$, and the transformed image $C_i^{Trans}$. While the transformations themselves were entirely random, they were sampled in a stratified manner, such that 44 pairs were considered to have undergone a "large" transformation, 45 a "medium" and another 45 a "small" transformation. An average displacement was considered small in not exceeding 100px, medium if of a size $(100, 200]$px, and large if exceeding 200px. The same evaluation metric was used for all registration results: $err = \frac{1}{4} \sum_{i=1}^{4} ||C_i^{Ref} - C_i^{Reg}||_2 > 100$px is considered a failure. $C_i^{Ref}, C_i^{Reg}, i \in \{1, 2, 3, 4\}$, are the corners of the reference image and the one resulting from the registration, respectively. As the ground truth was obtained by manual registration, we included an independent manual registration task on a subset of our experimental setup by 6 human annotators which showed that a displacement error of up to $\sim 50$ pixels can be expected; see App. 7.4, Fig. 7.

**Implementation details** We make no assumptions about the properties of the model. We choose two identical dense U-Nets [29], one per modality, which share no parameters. We experiment with both the MSE and cosine similarity as a critic. We set the temperature to $\tau = 0.5$. We use 46 negative samples. An important parameter to choose is the field of view of the patches that the model is trained on. This is data-specific and a patch should be large enough to cover significant structures in the images. For both datasets patches of the size $128 \times 128$px were chosen. For the Zurich dataset we choose 3-channel CoMIRs, for the biomedical dataset we choose 1-channel CoMIRs as this is suitable for the downstream registration methods considered. The SHG images were preprocessed by

applying a log-transform $\log(1 + x)$ with $x \in [0, 1]$. Implementation details of our method as well as the subsequent registration algorithms can be found in App. 7.3.

**Baseline** To set a baseline performance, we perform multimodal registration using Mattes MI algorithm [42] on the original SHG and BF images with a (1+1) evolutionary strategy [50].

**Intensity-based Registration using $\alpha$-AMD** [44] is a general registration method based on distances combining intensity and spatial information [38]. It has been shown to be both accurate and having a large convergence region. One of its limitations, compared to e.g. MI, is that it requires the intensities to be in $[0, 1]$ and approximately equal (not merely correlated) for corresponding structures.

**Feature-based Registration using SIFT** SIFT is a feature detection algorithm introduced by [40]. It extracts features from both a reference and a floating image which are invariant to scale and rotation, and robust across a large range of affine distortions, additive noises, and changes in illumination. The extracted feature points are matched with Random Sample Consensus (RANSAC [17]).

**Manual Registrations** To obtain a baseline for comparing the machine performance to a human level, a panel of six annotators was selected to perform the registration on a small set of test patches (n=10, randomly selected, identical for all annotators). The setup for this registration task is the same as for the automatic registration methods: $834 \times 834$px patches from the center of the tissue microarray cores are to be aligned. Note that this differs significantly from the setup in which the ground truth (GT) for the biomedical dataset was originally manually acquired where the full cores were available to the annotator as shown in App. 7.2, Fig. 6. The GT was obtained by manually aligning the selected landmarks [30], while the manual registrations in our evaluation is performed by moving the SHG images over the bright-field images to obtain an appropriate fit.

**Generative Methods** Generative models, such as generative adversarial networks (GANs), are often used in image-to-image translation ([28, 64]) in which they have the potential to enable the use of monomodal registration methods by translating one modality into the other. We implement three well-known image translation methods: **pix2pix** [28], **CycleGAN** [64], and **DRIT** [35, 36]. As image translation aims to predict a representation of a BF image given the SHG input and vice versa, the resulting images can be considered to be in a common space in which monomodal registration can be attempted.

**Data-specific State-of-the-Art** The first intensity-based registration method capable of automatically aligning SHG and BF images was proposed as CurveAlign in [30]. CurveAlign relies on data-specific, biomedical knowledge, i.e. that BF images are stained by hematoxylin and eosin (H&E), where eosin stains extracellular matrix components such as collagen, in particular with shades of pink. SHG mainly corresponds to collagen fibers. Using this prior on the data, CurveAlign performs segmentation on the BF image to isolate collagen structures, which are then registered to the SHG using a registration scheme based on MI with a $(1 + 1)$ evolutionary algorithm.

**Choice of Critic** The effect of the choice of critic can be visually inspected on an image from the Zurich test set, in App. 7.4, Fig. 11. The correlation between the CoMIRs of RGB and NIR, for each of the observed critics – bilinear model, cosine similarity, and MSE – was averaged across a set of temperature settings $\tau$ (N=15). The results show that a bilinear critic produces weakly correlated maps w.r.t. $\tau$: $\bar{\rho}_{\text{bilinear}} = 60.9\%[56.7\%; 65.0\%]$. Using MSE resulted in $\bar{\rho}_{\text{MSE}} = 85.4\%[83.2\%; 88.1\%]$, whereas cosine similarity gave $\bar{\rho}_{\text{cosine}} = 91.5\%[88.4\%; 94.7\%]$. The 95% Confidence Intervals (C.I.) were computed with empirical bootstrap. Hence, we perform the registration evaluation on the biomedical dataset for MSE and cosine similarity. Training with the MSE gave consistently better CoMIRs w.r.t. registration, see App. 7.4, Fig. 9.

**Results** Fig. 3 shows an image pair of the biomedical test set, together with the representations produced by the image translations utilizing pix2pix, CycleGAN and DRIT, as well as the corresponding CoMIRs. The arrows indicate the pairs of modalities for which the registration task was attempted. Using image pairs resulting from image translation led to poor performance using both intensity-based as well as feature-based registration. The similarities between the GAN generated images and their original counterparts appeared too large for registration by $\alpha$-AMD. The corresponding SIFT features were detected for only three image pairs among all combinations of image translations and modalities, but even for these the registration error exceeded our success-threshold of 100px. We report the results for registration by MI for these methods in App. 7.3 Fig. 8.

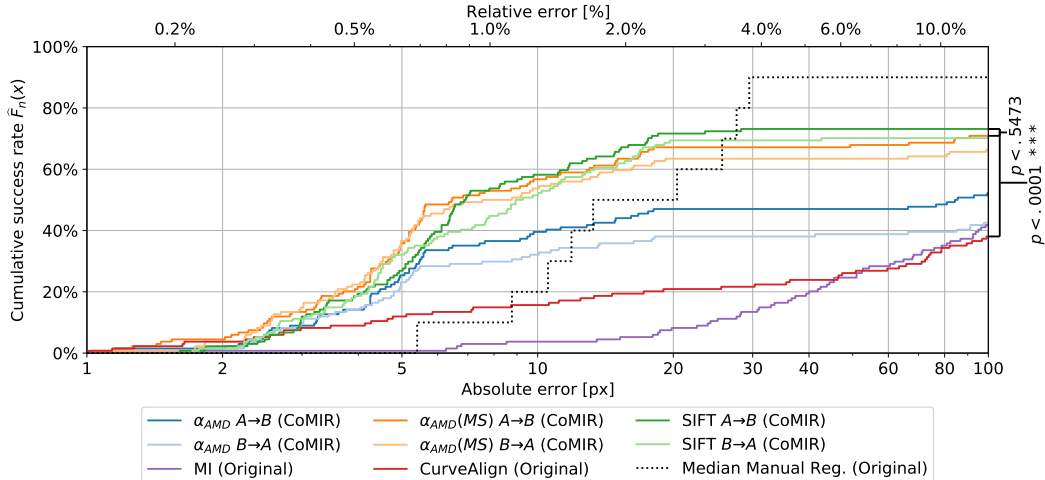

Figure 4: The cumulative number of successful registrations for the increasing error shown as an eCDF of the different registration methods over the biomedical test set. The results are compared to the median of the error of six independent manual registrations on a subset of ten images. The relative error is the proportion to the patch width and height. A Wilcoxon signed-rank test is performed to statistically highlight the differences between CurveAlign, SIFT, and $\alpha$-AMD.

Fig. 4 shows the empirical cumulative distribution functions (eCDF) of the successful registrations on the CoMIRs based on MSE, as a function of the error for (i) $\alpha$-AMD, (ii) $\alpha$-AMD with multiple starts (MS), and (iii) SIFT, using both CoMIR learnt from BF (A $\rightarrow$ B) and SHG (B $\rightarrow$ A) as the reference image. As a baseline comparison we also show the eCDF of (iv) the MI registration on the original BF and SHG images, (v) CurveAlign which registers BF to SHG, as well as (vi) the median eCDF for the six manual registrations on a subset of ten images. A registration error not smaller than 100 is considered a failure and is not shown. The feature- and intensity-based registration performances on CoMIRs are consistently better than the multimodal registration approaches on the original multimodal images. Wilcoxon signed-rank tests show that SIFT significantly ($p = 5e{-}11$) outperforms CurveAlign, but show no significant difference between $\alpha$-AMD MS and SIFT ($p < .5473$). The median error on the manual registration highlights the difficulty of the task. We further observe that the success of registration by MI is highly dependent on the size of the transformation between the image pair. We perform registration by MI (i) on the original images; (ii) for the corresponding CoMIRs; (iii) the BF and GAN generated BF as well as (iv) SHG and GAN generated SHG for pix2pix, CycleGAN and DRIT and observe linear dependence between the mean registration error and the mean displacement of the corner points for all attempts. The results are shown in App. 7.4, Fig. 8. The choice of critic influences the resulting CoMIRs. MSE encourages the intensities of the representations to be more similar than cosine similarity, which is favorable for the registration task. We observe that for both registration by SIFT, as well as $\alpha$-AMD, the results are considerably better for the MSE-based CoMIRs than for the cosine similarity, see App. 7.4 Fig. 9. App. 7.4 Table 2 gives the number of successfully registered image pairs which resulted in less than $1\%$ relative error to the image width and length ($< 9\text{px}$), as well as $5\%$ relative error ($< 42\text{px}$). Both the training (GPU, $1345\text{s}$) and inference (CPU, $5\text{s/image}$) of the model are fast, and the process of registration adds between 2s/image with SIFT and 25s/image with $\alpha$-AMD MS (see App. 7.4.1). An animated illustration depicting the process of registration of three example images with $\alpha$-AMD is available at: `https://youtu.be/zpcgnqcQgqM`.

## 5 Conclusion

The contrastive loss based on InfoNCE was successfully adjusted to the task of image registration and modified to result in rotationally equivariant features. The proposed CoMIRs successfully extract shared content in multimodal images to enable multimodal image registration by reducing it to a monomodal one. Using monomodal intensity- and feature-based registration methods significantly outperforms multimodal registration by MI, as well as a state-of-the-art, data-specific approach. For registration tasks, the generated CoMIRs contain more valuable information than GAN generated images. We show that the training of CoMIRs is stable w.r.t. hyperparameters and reproducible in

connection with weight initialization and training data. We show that the size of the training dataset can be as little as one image pair and give insight to the choice of critic. Future research could explore adding dense aleatoric uncertainty to CoMIRs, learning CoMIRs of more complex and higher dimensional data and more diverse modalities (e.g. volumetric images, audio, videos, and time-series) and extend equivariant properties to other groups, as well as investigate the CoMIRs' applicability to segmentation and pixel regression tasks.

## 6  Broader Impact

Using CoMIRs for multimodal image registration has a direct application in multimodal image fusion. A wide range of areas benefit from fusing content of images of different sensors. One such area is material science. Early stage anomaly detection is used to characterize newly developed materials w.r.t. physical properties. As a concrete example, carbides along the grain boundary of a material can indicate impairments in material strength, but require different Scanning Electron Microscopy (SEM) sensors which acquire images asynchronously at different spatial resolutions [7]. This results in a multimodal registration problem which could be addressed by the proposed CoMIRs. The implications research in material science based on successful registration and fusion of images of this kind can have on the society are widespread. The Materials Genome Initiative (MGI) which has been launched by the US Federal Government in 2011 for example, aims to address clean energy, national security, and human welfare [41]. While this area of research can have a beneficial impact on society, by developing biocompatible materials for medical advances or materials needed in a variety of settings to reduce the carbon footprint, scientific findings in this area are directly linked to military developments also (see e.g. the US Air Force's involvement in MGI). Another area of application which makes use of multimodal imaging data is the field of remote sensing. Again, while aerial observation can be used to monitor geological changes like melting glaciers due to climate change or early detection of wildfires, it is also tightly connected to military action (espionage, navigation systems such as Unmanned Aerial Vehicles (UAV), target localization, lethal autonomous war weapons). The area which can profit the most from multimodal image registration and following fusion in a positive way, is biomedicine. It is a broad and active field of research to combine information from modalities such as computed tomography (CT), magnetic resonance imaging (MRI) and Positron emission tomography (PET) or BF, SHG and two-photon-excited fluorescence (TPEF) microscopy. These imaging techniques often provide complementary and clinically relevant information needed for a diagnostic task. For example CT gives good spatial resolution and dense tissue contrast, while MRI yields better soft tissue contrast. BF and SHG are for example studied together in connection with collagen organization in tumor growth. By providing a joint representation between multiple modalities, issues regarding patient data anonymization need to be taken into account, e.g. if only images in one modality were subject to anonymization (e.g. photographs), while the other was not (e.g. CT scan), CoMIRs could potentially facilitate data de-anonymization from one modality to another. Registration by MI is computationally expensive, and as we show in our paper, MI is highly susceptible to an initial starting position close to the global extremum. In order to perform well, a larger number of restarts is needed to overcome being trapped in a local extremum, increasing the computational load even more. Generating CoMIRs for an entire dataset combined with registration by SIFT is much cheaper computationally, especially regarding the small training data needed for CoMIR generation. Hence, in settings where multimodal registration is already in place, CoMIRs can reduce the energy cost required and in consequence the environmental impact, however it may encourage the analyses of multimodal data on a much greater scale than ever before. The CoMIRs' usage is also not limited to the task of multimodal registration, but could be useful for classification, segmentation or patch retrieval. This is subject of future research, but we foresee a potential in segmentation by training on the label masks as a second modality for example.

### Acknowledgments and Disclosure of Funding

We would like to thank Prof. Kevin Eliceiri (Laboratory for Optical and Computational Instrumentation (LOCI), University of Wisconsin-Madison) and his team for their support and kindly providing the dataset of BF and SHG imaging of breast tissue microarray cores.

The project was financially supported by the Swedish Foundation for Strategic Research (grants SB16-0046, BD150008), the European Research Council (grant 682810), the Wallenberg Autonomous Systems and Software Program, WASP, AI-Math initiative, VINNOVA (MedTech4Health project 2017-02447) and the Swedish Research Council (project 2017-04385).

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
