[Supplementary Material]

# 7 Appendix

## 7.1 Algorithm

In this section, we present a general algorithm for (supervised) training of models for generating CoMIRs given a dataset of aligned images. The CL used in this work is not fundamentally limited to two modalities, but can be extended for $M$ modalities.

Let $\mathcal{D} = \{(\boldsymbol{x}_i^1, \ldots, \boldsymbol{x}_i^M)\}_{i=1}^n$ be an i.i.d. dataset containing $n$ data points, where $\boldsymbol{x}^j$ is an image in modality $j$, and $f_{\boldsymbol{\theta}_j}$ the network processing modality $j$ with respective parameters $\boldsymbol{\theta}_j$ for $j \in \{1, \ldots, M\}$. For an arbitrary datapoint $\boldsymbol{x}_i = (\boldsymbol{x}_i^1, \boldsymbol{x}_i^2, \ldots, \boldsymbol{x}_i^M) \in \mathcal{D}$, $i \in \{1, 2, \ldots, n\}$ the loss is given by

$$\mathcal{L}^{opt}(\mathcal{D}) = -\mathbb{E}_{\boldsymbol{x}^1, \cdots, \boldsymbol{x}^M \sim \mathcal{D}} \left[ \log \frac{\frac{p(\boldsymbol{x}^1, \cdots, \boldsymbol{x}^M)}{\prod_{i=1}^M p(\boldsymbol{x}^i)}}{\frac{p(\boldsymbol{x}^1, \cdots, \boldsymbol{x}^M)}{\prod_{i=1}^M p(\boldsymbol{x}^i)} + \sum_{\boldsymbol{x}_j \neq \boldsymbol{x}} \frac{p(\boldsymbol{x}_j^1, \cdots, \boldsymbol{x}_j^M)}{\prod_{i=1}^M p(\boldsymbol{x}_j^i)}} \right] \tag{5}$$

The pseudo-code for training models using the CL to produce CoMIRs, based on aligned images, is given in algorithm 1.

---

**Algorithm 1** CoMIR learning algorithm
---

**input:** batch size $N$, set of datapoints $\{\boldsymbol{x}_i\}_{i=1}^N$ in $M$ modalities, a group $\mathcal{G}$ (e.g. $\mathcal{C}_4$), a critic function $h(\cdot, \cdot)$, a temperature $\tau$.
initialize all models $\{f_{\boldsymbol{\theta}_i}\}_{i=1}^M$
**for** sampled mini-batch $\boldsymbol{x} = \{\boldsymbol{x}_k\}_{k=1}^N$ **do**
    # Computation of the latent spaces
    **for all** $m \in \{1, \ldots, M\}$ **do** # for each modality
        **for all** $k \in \{1, \ldots, N\}$ **do** # for each element in batch
            draw an operation $T \sim \mathcal{G}$
            $\boldsymbol{y}_k^m = T'(f_{\boldsymbol{\theta_m}}(T(\boldsymbol{x}_k^m)))$
        **end for**
    **end for**
    # Computation of the similarity matrix
    $\boldsymbol{z} = \|_{i=1}^M \boldsymbol{y}^i$ # Concatenated latent spaces $(MN \times 1)$
    $\boldsymbol{S} \in \mathbb{R}^{MN \times MN}$
    **for all** $i \in \{1, \ldots, MN\}$ **do**
        **for all** $j \in \{1, \ldots, i\}$ **do**
            $\boldsymbol{S}_{i,j} = \boldsymbol{S}_{j,i} = \log h(\boldsymbol{z}_i, \boldsymbol{z}_j) - \log \tau$
        **end for**
    **end for**
    # Computation of the loss
    $\mathcal{L} = 0$
    **for all** $i \in \{1, \ldots, MN\}$ **do**
        **for all** $m \in \{2, \ldots, M\}$ **do**
            $\mathcal{L} = \mathcal{L} - \boldsymbol{S}_{(i,mN+i)\%MN} + \log \left( \sum_{j=1}^{MN} e^{\boldsymbol{S}_{i,j}} - \sum_{j=1}^M e^{\boldsymbol{S}_{(i,jN+i)\%MN}} + e^{\boldsymbol{S}_{(i,mN+i)\%MN}} \right)$
        **end for**
    **end for**
    $\mathcal{L} = \frac{1}{MN(M-1)} \mathcal{L}$
    update all models $\{f_{\boldsymbol{\theta}_i}\}_{i=1}^M$ to minimize $\mathcal{L}$
**end for**
**return** all models $\{f_{\boldsymbol{\theta}_i}\}_{i=1}^M$

---

## 7.2 Datasets

**Zurich Dataset** The dataset can be downloaded at https://sites.google.com/site/michelevolpiresearch/data/zurich-dataset. Each image was acquired with QuickBird and consists of four channels in the order [NIR,B,G,R]. An example image of the dataset is given in Fig. 5. In all our experiments we train on only one image of the Zurich zh10.tif, except for the experiments in table 1 for which we investigate the impact of the training distribution and vary which image from the Zurich dataset is used as the training image. As a test image to evaluate the representations zh12.tif was chosen.

Figure 5: Example image of the Zurich dataset. To the left the RGB image, to the right the corresponding NIR image.

**Biomedical Dataset** The dataset used in this study was kindly provided by the authors of [30] and modified for this study. The modified dataset used in our experimental setup can be downloaded at `https://zenodo.org/record/3874362`, [16]. Multimodal Image Registration is of particular interest in biomedical tasks such as the registration of bright-field (BF) microscopy images and second-harmonic generation (SHG). The two ways of imaging provide complementary information and are of particular interest in studying collagen organization in cancerous tissue, [3], [15] [14] and [4]. In general the two modalities are often captured in different microscopes which means that the SHG sample must be relocated and registered within a BF scan. This is a challenging task as the modalities result in very different signals and have little appearance in common. The difficulties of this registration task are discussed in detail in [30], which provides the first intensity-based registration method, called CurveAlign, capable of automatically aligning SHG images and BF images that are usually aligned manually. They evaluate their methods on tissue microarray cores as can be seen in Fig. 6 to the left. In this paper we use center-cropped patches within these cores, marked by the green square. This registration task is harder as the overview of larger structures and the tissue boundary are missing. Using these patches allows us to avoid any padding effects after applying random transformations as described in 4.2. The GT alignment used for the evaluation is based on manual landmark annotation provided by [30] and was done on the full tissue microarray cores. The manual registration performed as part of the evaluation in this paper is however performed on the cropped patches and by overlaying the SHG image on to the BF image. The two kinds of manual registrations (GT and manual registration in the evaluation) are hence not one to one comparable. The manual registration in the evaluation was done according to the setup used in the automated methods, to give a baseline and sense of the magnitude of pixel error that can be expected in this task.

By enabling registration on patches within the tissue micro arrays, a further downstream task of patch retrieval can be achieved. This is in particular interesting for finding SHG patches within BF whole slide images as is relevant in ongoing research such as [27].

## 7.3 Implementation Details

We chose the same dense U-Net architectures for experiments on the Zurich and the biomedical dataset [29], using 32 convolutional filters for the first convolution, 4 dense blocks of depth 6 as down and up blocks and 4 bottleneck layers. Upsampling was used to avoid grid artifacts. Max pooling, a dropout rate of 0.2, no early transition or activation function in the last layer and a compression rate of the convolutional layers of 0.75 were used. The commonly used non-linear activation in the final layer is omitted. The 1-channel CoMIRs illustrated in 3 are visualized by applying a logistic function with temperature of 0.5. Illustrations done on the Zurich dataset are made by normalizing two corresponding CoMIRs jointly by the maximum of the 1-percentile and the minimum of the 99-percentile of the two representations.

**Experiments on Zurich Dataset** For generating CoMIRs in experiments on the Zurich dataset, Adam optimizer was used with a learning rate of $1e-3$, a weight decay of $1e-4$. Temperature $\tau$ was set to 0.1, the batch size to 24, and the steps per epoch to 32. The gradient norm was limited to 1. $L_1$ and $L_2$ activation decay were set to 0. To generate a batch, patches of size $128 \times 128$ were cropped from the training image. Data augmentation consisted of flips ($p = 0.5$) and random rotations from by up to $\pm 180°$ using either a linear, nearest neighbor or cubic interpolation randomly. With a probability $p = 0.2$ either additive Gaussian noise ($\mu = 0$, $\sigma \in (0., 0.05)$ randomly chosen), Gaussian blur ($\sigma = 0.1$), or coarse dropout (rate of dropout of 10% per channel, superpixel

Figure 6: This example image pair of the test set shows the different usage of the data introduced and registered in [30]. The registration problem becomes harder by using the cropped patches marked in green, as larger structures that can aid the registration are not available. The GT of the dataset as obtained in [30] by a manual user choosing landmarks was performed on the large images to the left, while the manual registrations in our evaluation are performed on the patches to the right as described in 4.2.

Figure 7: Variation in manual annotation errors across test patches (blue dots represent errors of individual annotators) compared to the best performing registration methods, $\alpha_{AMD}$ and SIFT, applied on CoMIRs. The parameters $\hat{\sigma}$ are estimated along with their 95% confidence interval (Rayleigh distribution).

size of 5%) is applied or the edge image is taken. Lastly, each channel is multiplied randomly (p=0.3) by a value in $(0.9, 1.1)$. The models were trained for 5 epochs.

**Experiments on Biomedical Dataset** For generating 1-channel CoMIRs for the multimodal registration experiments, stochastic gradient descent was used with a learning rate of $1e-2$, a weight decay of $1e-5$ and a momentum of $0.9$. Temperature $\tau$ was set to $0.5$, the batch size to 32, and the steps per epoch to 32. The gradient norm was limited to 1. $L_1$ activation decay was set to $1e-4$, $L_2$ activation decay to $1e-4$. To generate a batch, patches of size $128 \times 128$ were cropped from the training image and the data augmentation consisted of flips ($p = 0.5$) and random integer rotations by up to $\pm 180°$ using either a linear, nearest neighbor or cubic interpolation randomly. The models were trained for 23 epochs in the efficient mode of the model implementation.

**Registration using $\alpha$-AMD** For $\alpha$-AMD, 3 resolution levels were used with sub-sampling factors given by $4, 2, 1$ and Gaussian smoothing $\sigma$ given by $12, 5, 1$ respectively. A SGD optimizer was used with a momentum of $0.9$, and step-sizes $(2, 2, 2 \rightarrow 0.2)$ where the transition in the last resolution level is a linear interpolation between 2 and 0.2, and a hard gradient clipping at a magnitude of 1, parameter-wise. The logistic function (without temperature) is applied to the CoMIRs to map them into the range $[0, 1]$, which is needed for the method, and then the images were quantized into 7 (non-zero) levels. The number of iterations, per resolution level, are $3000, 1000, 500$ respectively. A sampling fraction of $0.005$ was used. For the multistart version of the method, three starts were chosen at rotations (in radians) in $\{-0.3, 0, 0.3\}$, and the registration with the lowest final distance value was chosen as the output of the algorithm. We used a modified version of an open-source implementation (https://github.com/MIDA-group/py_alpha_amd_release) of the method. We provide the modified version in the github repository.

**Registration using SIFT** The SIFT implementation in Fiji 2.0.0 was used based on the `mpicbg.imagefeatures` package. The feature descriptor size was set to 4 samples per row and column, the orientation bins for 8 bins per local histogram. The scale octaves were set to be in $[128, 1024]$px with 3 steps per scale octave and an initial $\sigma$ of each scale octave equal to $1.6$.

**Registration by Mutual Information** Registration by Mattes MI was implemented in Matlab 2019b using a (1+1) evolutionary algorithm with an initial size of the search radius of 1e-5, a minimum size of the search radius of 1.5e-8, a growth factor of the search radius of (1+1e-4) and a maximum number of iterations of 1500. The number of spatial samples for the MI computation was 500 and the number of histogram bins 80. All pixels were included in the overlap region. As registration by MI required 1-channel input, all 3-channel representations (BF and GAN generated BF images) have been reduced to one channel by principal component analysis (PCA).

**Registration by CurveAlign** CurveAlign v5.0 Beta provides HSV color based registration of SHG and BF images. The method was run with the default settings, however it was set to rigid registration, where the default is affine. The code is provided in `https://github.com/uw-loci/curvelets`.

**Manual Registration** The registration was performed using in Fiji 2.0.0 with TrakEM 2. The mean error of the corners after manual registration for ten randomly chosen images of the test set is shown in Fig. 7.

Table 2: Number of successful registrations on the test set (N=134) for multiscale $\alpha$-AMD, $\alpha$-AMD, SIFT, MI and CurveAlign which resulted in less than 1% pixel error ($< 9$px) and less than 5% pixel error ($< 42$px). $\widehat{BF}$ and $\widehat{SHG}$ denote the fake images produced by a GAN image translation given the respective other modality. 95% C.I. are Clopper-Pearson intervals.

| Reg. Method | | MS $\alpha$-AMD | | $\alpha$-AMD | | SIFT | | MI | | CurveAlign |
|---|---|---|---|---|---|---|---|---|---|---|
| **Originals** | Input | — | — | — | — | — | — | BF→SHG | SHG→BF | BF→SHG |
| | Succ.($< 1\%$ Err.) | — | — | — | — | — | — | 7 [3; 14] | 7 [3; 14] | **22** [14; 32] |
| | Succ.($< 5\%$ Err.) | — | — | — | — | — | — | 30 [21; 41] | 29 [20; 40] | **33** [24; 44] |
| **CoMIR MSE** | Input | A→B | B→A | A→B | B→A | A→B | B→A | A→B | B→A | — |
| | Succ.($< 1\%$ Err.) | 72 [60; 84] | 70 [58; 82] | 49 [38; 61] | 43 [33; 55] | **75** [63; 86] | 67 [55; 79] | 7 [3; 14] | 9 [4; 17] | — |
| | Succ.($< 5\%$ Err.) | 90 [78; 101] | 87 [75; 98] | 63 [51; 75] | 54 [43; 66] | 98 [87; 108] | 93 [82; 103] | 30 [21; 41] | 30 [21; 41] | — |
| **CoMIR Cos.** | Input | A→B | B→A | A→B | B→A | A→B | B→A | A→B | B→A | — |
| | Succ.($< 1\%$ Err.) | **54** [43; 66] | 53 [42; 65] | 19 [12; 28] | 18 [11; 27] | 47 [36; 59] | 52 [41; 64] | 12 [6; 20] | 11 [6; 19] | — |
| | Succ.($< 5\%$ Err.) | 62 [50; 74] | 62 [50; 74] | 34 [24; 45] | 33 [24; 44] | 67 [55; 79] | **69** [57; 81] | 29 [20; 40] | 27 [18; 37] | — |
| **CycleGAN** | Input | $\widehat{BF}$→BF | $\widehat{SHG}$→SHG | $\widehat{BF}$→BF | $\widehat{SHG}$→SHG | $\widehat{BF}$→BF | $\widehat{SHG}$→SHG | $\widehat{BF}$→BF | $\widehat{SHG}$→SHG | — |
| | Succ.($< 1\%$ Err.) | — | — | — | — | 0 [0; 4] | 0 [0; 4] | **6** [2; 13] | 5 [2; 11] | — |
| | Succ.($< 5\%$ Err.) | — | — | — | — | 0 [0; 4] | 0 [0; 4] | **30** [21; 41] | 28 [19; 39] | — |
| **Pix2Pix** | Input | $\widehat{BF}$→BF | $\widehat{SHG}$→SHG | $\widehat{BF}$→BF | $\widehat{SHG}$→SHG | $\widehat{BF}$→BF | $\widehat{SHG}$→SHG | $\widehat{BF}$→BF | $\widehat{SHG}$→SHG | — |
| | Succ.($< 1\%$ Err.) | — | — | — | — | 0 [0; 4] | 0 [0; 4] | 6 [2; 13] | **7** [3; 14] | — |
| | Succ.($< 5\%$ Err.) | — | — | — | — | 0 [0; 4] | 0 [0; 4] | 29 [20; 40] | **30** [21; 41] | — |
| **DRIT** | Input | $\widehat{BF}$→BF | $\widehat{SHG}$→SHG | $\widehat{BF}$→BF | $\widehat{SHG}$→SHG | $\widehat{BF}$→BF | $\widehat{SHG}$→SHG | $\widehat{BF}$→BF | $\widehat{SHG}$→SHG | — |
| | Succ.($< 1\%$ Err.) | — | — | — | — | 0 [0; 4] | 0 [0; 4] | 5 [2; 11] | **6** [2; 13] | — |
| | Succ.($< 5\%$ Err.) | — | — | — | — | 0 [0; 4] | 0 [0; 4] | **31** [22; 42] | 29 [20; 40] | — |

**Alternative methods** pix2pix, CycleGAN and DRIT train on image patches created and augmented in the same way as for the gernation of CoMIRs, however the patches are not created during runtime but before training. The code for the competing methods are provided in `https://github.com/junyanz/pytorch-CycleGAN-and-pix2pix` for pix2pix and CycleGAN and `https://github.com/HsinYingLee/DRIT` for DRIT.

The pix2pix and CycleGAN models were trained for 100 epochs with the initial learning rate of $2e-4$ and another 100 epochs with a linearly decaying learning rate thereafter. Adam optimizer was used with $\beta_1 = 0.5$. The GAN-objective was `lsgan`, the discriminator architecture was a $70 \times 70$ PatchGAN and a 9-block ResNet for the generator architecture. 64 filters were used in the last convolutional layer of the generator as well as the first generator of the discriminator. Instance normalization was used and no dropout for the generator. During the test phase, the input image sizes were padded to minimum multiples of 256 (i.e. as the size of input images were 834, they were padded to 1024) due to the network architectures' restriction for pix2pix [28] and CycleGAN [64]. To avoid edge artifacts during inference, the padded borders were filled with the reflection of the vector mirrored on the first and last values of the vector along each axis. The padded areas of translated images are cropped off before further processing.

DRIT [35, 36] was adjusted to fit the registration task, by using the disentangled attribute representation of the image for registration with the corresponding image, rather than using the disentangled attribute representation of a random given image from the other modality. The discriminator had no normalization layers and its scale was set to 3. The model was trained for 1200 epochs, with a learning rate decay for the last 600 epochs.

## 7.4 Results

In table 2 the number of successfully registered image pairs which resulted in less than $1\%$ relative error to the image width and length ($< 9$px), as well as $5\%$ relative error ($< 42$px) is given for each applicable registration method. Fig. 7 shows the results of the manual registration on ten image pairs from the test set together with the result of the automated registration based on $\alpha$-AMD and SIFT. Fig. 8 shows the error of registration results based on MI with respect to the displacement of the corner points resulting from the transformation applied to the image. To point out the importance of the extent of the transformation which needs to be recovered in case of MI, SIFT results are also shown for comparison. Fig. 9 shows the eCDF of automatic registrations of the biomedical test set on the CoMIRs learnt by using a critic based on MSE and cosine similarity. Fig. 10 shows the influence weight initialization and a particular training image have on the CoMIRs (3 channels) generated of an image patch of image `zh12.tif` in the Zurich test set. Fig. 11 shows the influence of temperature $\tau$ and the choice of critic on the generated CoMIRs (3-channel) of `zh12.tif`.

### 7.4.1 Time Analysis

The experiments are run on an Intel(R) Core(TM) i9-7980XE CPU @ 2.60GHz (hyperthreading enabled) and a Titan V (12GB) GPU. Table 3 shows the approximate time needed for registration using MI on the CPU versus generating CoMIR and registration using SIFT. The inference includes the loading of the dataset and the time reported includes generation and registration using each modality as a reference image (A → B and B → A).

Figure 8: The mean registration error of the corner points vs. the mean displacement of corner points is shown for registration by MI as well as SIFT. It can be seen that MI suffers from large transformations and fails to detect a global extremum if the initial starting position of the optimization is too far away. This depedence of the accuracy on the starting position can be observed for all modalities. In contrast, SIFT does not depend on the extent of the transformation between the images that have to be registered, as can be seen to the right.

Table 3: Approximate time for steps in registration pipelines given in seconds for the entire test set.

| Device | | Preprocessing | Registration using MI (Matlab) | Preprocessing + Registration |
|---|---|---|---|---|
| CPU 1 worker | | 114 | 9593 | 9707 |
| CPU 18 worker | | — | 896 | 1010 |
| | Training 23 epochs (Pytorch/Python) | Inference (Pytorch/Python) | Registration using SIFT (Fiji/Java) | Inference + Registration |
| GPU | 1345 | 49 | — | 324 |
| CPU 1 worker | — | 2631 | 275 | 2906 |
| CPU 18 worker | — | 1020 | — | 1295 |

While registration using MI will scale linearly with the number of image pairs which need to be registered, the training of the proposed model can be considered constant overhead and the generation of CoMIRs takes only 0.2 seconds per image on a GPU. After training the inference can also be done on the CPU, albeit is slower, but can be beneficial in a clinical setting where no access to a GPU can be given, or to encode images of large spatial dimensions which do not fit onto a GPU. It should be noted that no parallelization attempt was made for registration by SIFT nor for the preprocessing (turning the images to 1-channel images by PCA), which could speed up the registration further.

Figure 9: eCDF of the automatic registration methods over the biomedical test dataset for CoMIRs produced by the MSE-based critic as well as the cosine similarity-based critic ((a) $\alpha$-AMD, (b) $\alpha$-AMD MS, (c) SIFT and (d) MI). Wilcoxon tests between the results based on MSE and cosine similarity are shown on the right side of each plot, along with the matched-pairs rank-biserial correlation $r$ (effect size)[31].

(a) CoMIRs of models trained with different weight initializations and different images.

(b) CoMIRs of models trained with different weight initializations and a unique common image.

(c) CoMIRs of models trained with the same weight initialization and different images.

(d) CoMIRs of models trained with the same weight initialization and a unique common image.

Figure 10: Different experiments were performed to show that a random weight initialization scheme with a fixed seed yields similar models. Using a small training set increases the similarity of the trained models. All models were trained on only one training image, either fixed or varying, depending on the experiment. (a) and (c) have the 12th image omitted as they were trained on the same image as the test image, used to create this figure.

(a) CoMIRs of models trained with the MSE based critic.

(b) CoMIRs of models trained with the cosine similarity based critic.

(c) CoMIRs of models trained with the bilinear model based critic.

Figure 11: CoMIRs generated from models trained with different losses and temperatures $\tau$. Top-rows are generated with modality 1 (RGB), bottom rows with modality 2 (NIR). A high temperature makes the CoMIRs more blurred. The models were initialized with the same seed.