[Reviews · NeurIPS 2020]

Review 1

Summary and Contributions: The author's motivation is to perform image registration between two different modalities. They utilize contrastive learning for the regression task and apply data augmentations to achieve rotational equivariance.

Strengths: The main contribution might be the MSE critic. The problem could be considered as contrastive regression learning which is relevant to the Supervised Contrastive Learning by P. Khosla et al., 2020

Weaknesses: Image registration for rotated images is a simple case, while diffeomorphic transformation is more challenging and close to real-life problems. While the free-form transformations are ill-conditioned and hard to regularize, rotation is a trivial assumption to represent the actual deformation (i.e. in tissues). The computational complexity of the approach for N modalities is O(N^2). The authors can explore ideas from the generalized canonical correlation analysis (gCCA) (Paul Horst et al.). In the case of extension to volumetric data, the data-augmentation approach for rotational-equivariance would be much more expensive. There is little discussion into why cosine similarity would fail compared to the MSE critic given that it has a higher correlation. While bilinear critic was excluded due to smaller correlation.

Correctness: The claims and empirical methodology should be strengthened according to their weaknesses.

Clarity: The paper is generally well-written and structured clearly. However, it was hard to distinguish in the results section which critic was used with CoMIR. Specifically, Table 1 probably used cosine similarity, while in Figure 4 the results are shown with MSE critic after searching it in the paragraph.

Relation to Prior Work: The authors clearly discussed the difference from previous contributions. However, authors didn't mention other cases of the image registration problem (for example, deformable models or work done by medical imaging community).

Reproducibility: Yes

Additional Feedback: Authors have explored only a simple case of the image registration problem by applying recent work on contrastive learning with augmentation techniques. I would strongly consider this manuscript as a mostly application paper since there is a lack of novelty in the method itself. It can be more compelling to the audience in the conferences for computer vision and medical imaging applications. Update: I want to thank the authors for their effort. I am considering most of my concerns as future directions. It is a good application of a data-augmented contrastive learning framework. However, I am concerned with the theoretical novelty of this work. I decided to increase the score.


Review 2

Summary and Contributions: This work uses a contrastive loss based on InfoNCE to obtain common image representations for multimodal image registration. The approach is tested on two different datasets (aerial images of Zurich and microscopy images) and shows excellent results over a number of tested baseline approaches. The manuscript is mostly very well written and the tackled application is highly relevant in many application domains. As minor points for improvements, some of the notations and explanations could be improved (see details below) and it would be nice to acknowledge related pre deep-learning work for multimodal registration.

Strengths: 1. Nicely written paper that is easy to read. 2. The proposed approach addresses the relevant problem of multimodal image registration and evaluates its performance on two different datasets. 3. The approach nicely builds on contrastive coding to obtain shared image representations for multi-modal images which can then be used to compute registrations. 4. The presented results compare very favorably to existing approaches.

Weaknesses: 1. Some of the notation is a bit unclear (see below). 2. While the approach is likely compatible with many different spatial transformation models it was not clear to me from the manuscript (or maybe I missed it somewhere) what transformation model was considered for the experiments. Is this rigid or affine or something more complex? This should be made clearer as it is a core choice for a registration algorithm. 3. line 274: I found the results section a bit confusing to read. Looking at the caption of Fig. 4 made it clearer to me what is compared with what and what registration approaches are being used (i.e., AMD for all no feature-point registrations). This could likely be explained a little clearer in the text itself. 4. It would be nice if the background section could acknowledge at least some of the pre deep-learning (DL) approaches that tried to tackle similar problems. As I assume that the more modern DL approaches likely outperform them, a direct comparison is not that necessary in my opinion, but I think it would be good to give credit to some of this earlier work. Two papers (not necessarily representative, there might be others that might be preferable to cite) that come to mind are: a) Wachinger, Christian, and Nassir Navab. "Entropy and Laplacian images: Structural representations for multi-modal registration." Medical image analysis 16.1 (2012): 1-17; b) Cao, T., Zach, C., Modla, S., Powell, D., Czymmek, K., & Niethammer, M. (2014). Multi-modal registration for correlative microscopy using image analogies. Medical image analysis, 18(6), 914-926. 5. It would be useful to state a bit clearer in the manuscript what the differences/similarities to InfoNCE are in the way it is used in this submission.

Correctness: yes.

Clarity: Overall, this is clearly written paper that is a pleasure to read. Though there are some small aspects that should be clarified (see details below).

Relation to Prior Work: Yes, the relations are mostly clear. Some pre deep-learning work should likely also be acknowledged (see comments above).

Reproducibility: Yes

Additional Feedback: - Eq. 2: It is not clear from this equation what the notation x_i\neq x in the sum means and what the second term really sums over. - line 128: I assume that y^1 and y^2 are the Comirs, correct? This should be clearly defined somewhere. - Eq. 3: The dependencies for Eq. 3 should be made more explicit, i.e., what the quantities are that this loss is minimized with respect to (the parameters of the f^1 and f^2 networks I believe. - line 131: It is not clear to me from the description why the MI bound gets tighter for n to \infty. Doesn't the log(n) term go to infinity as well then? - line 141 / Rotational equivariance section: Why is C4 an appropriate symmetry group here? Can these images not have arbitrary rotations? It is also not clear to me what Eq. 4 does and how it is used in the model. Why does Eq. 4 subsume all the symmetry constraints? - line 158: I cannot clearly follow the patch pairings. Where does 2n-2 come from? - Fig. 4: What pixel errors are acceptable for the targeted applications? After the rebuttal: I found the authors' rebuttal quite good and it addressed my concerns. So I am still advocating for accepting this paper.


Review 3

Summary and Contributions: This paper studies contrastive learning approaches to learn image registration where the goal is to match images expressed in different views. Their approach enables the registration of multimodal images where existing registration methods often fail due to a lack of sufficiently similar image structures and is derived via an extension of InfoNCE to enforce rotational equivariance of the learnt representations. The authors evaluate the learnt representations through registration of a biomedical dataset of bright-field and second-harmonic generation microscopy images and show that the proposed approach outperforms existing approaches.

Strengths: 1. This paper offers a nice perspective on incorporating equivariance in multimodal registration problems using a contrastive learning approach. The approach is a simple extension of existing contrastive learning by augmenting the positive samples with the invariances that one aims to capture. This is quite intuitive to me and I think this is a nice contribution. 2. The authors back up their algorithm with a nice experimental setup which is reasonably extensive spanning 2 datasets: one existing public dataset and one that they collected privately in the healthcare domain. 3. While i commend the authors for the experimental setup, there are some issues with the experimental results and reporting, see weaknesses below.

Weaknesses: 1. The reporting of results could be improved: the differences in figure 4 are quite small so it would help to run the experiments multiple times and reporting mean and standard deviation to get a better sense of the significance of the results. 2. Clarity can be improved in the experiments section: it could be made clearer which datasets are used for which experiment, especially Figure 4 - it seems like quantitative results are only being reported for 1 out of the 2 datasest being tested. 3. The only comparison was to the baseline MI estimator, but there's quite a lot of work in contrastive learning that can also be compared to. 4. Since the rotational equivariance is the main contribution of the paper it becomes quite important to have sufficient ablation studies and in-depth analysis of this component of the model - what is the best way to augment the data? Does more augmentation beyond the multiples of 90 degrees help even more? What if I care about other types of equivariance beyond rotational e.g. spatial, translation etc. ================Post Rebuttal================ I thank the authors for the effort they have put into the rebuttal. To my understanding the main contribution of the paper is a simple extension of existing contrastive learning (infoNCE objective) by augmenting the positive samples with the invariances that one aims to capture (e.g. 90 degree rotation). While this is quite intuitive to me and I think this is a nice contribution, I am still concerned about the novelty of this work in the context of a neurips paper. The other reviewers have also brought up this point. e.g. R2: 'It would be useful to state a bit clearer in the manuscript what the differences/similarities to InfoNCE are in the way it is used in this submission.' Since this is the main technical novelty in this paper, unfortunately I still do not believe that the authors have made this sufficiently clear. I think the paper does deserve credit for getting these methods working on several new tasks and datasets (aerial images of Zurich and microscopy images) and I find their experimental setup, evaluation, and discussion quite solid. However, depending on where one draws the line with respect to related work in other application domains, there might be significant or little novelty here - I believe that the closest application would be in using contrastive objectives for multimodal retrieval/multimodal alignment/cross-lingual alignment (e.g. https://arxiv.org/abs/1906.05849). Given these issues I am inclined to keep my score of weak reject.

Correctness: 1. The proposed method is quite intuitive and can be seen as augmenting the positive samples with the invariances that one aims to capture. 2. The methodology is also quite solid. It would be even better if the private medical dataset can be publicly released for future research in this direction. 3. There lacks sufficient quantitative results for me to be convinced of the utility of the proposed approach.

Clarity: 1. A lot more details can be added to the writing of the experiments section, making it clear what datasets are being used and which modalities are being registered. 2. The figures are nice and help the reader in understanding the paper.

Relation to Prior Work: Yes, discussion of related work is sufficient.

Reproducibility: Yes

Additional Feedback:


Review 4

Summary and Contributions: The paper proposes a contrastive learning approach to biomedical image registration. More specifically, the paper aims to learn so-called CoMIRs (Contrastive Multimodal Image Representations) so that the multimodal registration problem can be reduced to a monomodal registration problem in which general intensity based registration algorithms can be applied.

Strengths: - The proposed contrastive learning approach is interesting and novel. Given the general nature of the approach, this paper is relevant to the NeurIPS community. - The reported results demonstrate good performance on the datasets which have been used for testing

Weaknesses: - The proposed approach only considers rigid registration, no application to non-rigid/deformable registration is shown. - The proposed approach requires pairs of registered images. This is a very significant disadvantage as accurate registration for training is very challenging and sometimes impossible. This also means that this approach cannot be easily extended to non-rigid/deformable registration. Conventional approaches such as MI do not have this limitation (even some GAN-based approaches can be trained using unpaired data) - There are far too many reference to the appendix.

Correctness: The method seems well justified. Throughout the paper, it should be made more clear that this is a supervised approach (where registered pairs of images are required during training). This seems to be "hidden away" in the paper.

Clarity: Generally the paper is well written. However, there are several aspects of the paper which are very difficult to follow. In particular, the mathematical notation is not always easy to follow due to a lack of explanation of variables. Furthermore, there are far too many reference to the appendix, in particular in the results. This makes the paper hard to read as a self-contained manuscript.

Relation to Prior Work: Prior work is well referenced and differences to prior work is also well explained.

Reproducibility: Yes

Additional Feedback: Thank you for the rebuttal, I appreciate the clarifications, but some of the concerns remain.

[Author Response · NeurIPS 2020]

We thank the reviewers for their thorough evaluation. We believe that the reviews will help us to clarify and improve the quality of the submitted paper.

Our main contribution is to use contrastive learning for creating image-like embeddings suitable for registration, and modifying the InfoNCE loss to obtain the rotational equivariance property. This framework can be used with any critic, and we empirically show that MSE leads to good results for the registration task in particular. InfoNCE has been previously used to learn embeddings used in classfication and segmentation tasks in which the resulting subspace is required to feature properties such as separability between classes. However, we are the first ones to produce image-like, contrastive representations that possess the necessary equivariant properties to find a transformation (through classical registration methods, tested with rigid models) between the original inputs.

Our method requires aligned pairs of images available for training, but the registration algorithm applied to the CoMIRs can be chosen independently and could use affine or deformable models. We test rigid models in this study and outperform [29], the SOTA of intensity-based, affine registration based on biological properties between the very dissimilar modalities SHG and BF. While registration under diffeomorphic transformations is a very interesting problem to address on top of our current experiments, our comparison with the baseline and SOTA method [29] shows that rigid registration using rotations and translations is a very challenging task on this biomedical multimodal dataset (see Fig. 4), due to the little correlation between the modalities. Apart from comparing with [29], we also mention [22] and [25] as other works of the medical community. We acknowledge MIND, which is one of the pre deep-learning approaches, and use mutual information (non learning-based method) as a baseline. The proposed references seem relevant to our work, and could be added after an in-depth reading.

We show that data augmentation alone does not achieve rotational equivariance, but that equivariance is reached by modifying the loss (Fig. 2, augmentation is applied to both $C_4$ enabled and disabled experiments). The cost of data augmentation and equivariance enforcing components are independent of the number of dimensions; in 2D, for every step we sample one rotation from a set of four (0, 90, 180, 270 deg.), in 3D, we would still sample only one rotation but from a set of 24 rotations. The pipeline may have to train longer for the networks to experience the CoMIRs rotated in all possible directions sufficiently many times, but considering the observed fast convergence (23 epochs or  22 minutes [l. 566 & App. Table 3] for the biomedical dataset) we believe this is feasible. We show that multiples of $90°$ provide nearly perfect rotational equivariance (Fig. 2 & https://youtu.be/SRISQFfOVI4) where we ablate the component enabling the $C_4$ equivariance. It is reasonable to expect that the minute drops in performance around multiples of $45°$(Fig. 2, polar plot) disappear by using the $C_8$ group, but this relatively small improvement comes at the cost of interpolating the CoMIRs for both the forward and backward passes.

The reviews contain many suggestions on how to clarify and improve the article. They are all very relevant, and we appreciate and fully agree with those comments.

**Reviewer 1:** The main computational cost of the method is linear w.r.t. modalities, as only one model is trained per modality. However, some components of the loss grow quadratically (the similarity matrix is $MN \times MN$, with $M$ modalities and batch size of $N$) but are inexpensive to compute, in comparison to a forward pass of a model. As the other modalities are being used as negatives, doubling the number of modalities and dividing the batch size by 2 results in the same computational cost. We consider the critic function as a hyperparameter, which should be chosen according to the task at hand [53] and can differ from cosine similarity, as usually chosen for unsupervised classification. We hypothesize that the MSE encourages the intensities of the representations to be more similar than cosine similarity, which is favorable for registration. In the future, we would like to further explore and understand the effect of the choice of critic. We thank the reviewer for advising us to explore gCCA, which seems highly relevant.

**Reviewer 2:** We are thankful for the detailed comments regarding notations and will clarify the equations and text accordingly. The details regarding the MI bound for $n \to \infty$ are given in [45], App. A1. The question regarding what pixel error is acceptable is highly relevant. As the ground truth was obtained by manual registration, we included an independent manual registration task of a subset of our experimental setup by 6 human annotators which showed that a pixel error up to $\sim 50$ pixels can be expected (see App. Fig. 7).

**Reviewer 3:** Both datasets are publicly available for the community ([1] & [56]), to reproduce and compare our method and results. The best performing registration method using CoMIRs has a 3 times higher success rate, resulting in less than 1% pixel error, than the SOTA on the given modalities (SHG and BF, not using CoMIRs) and more than double the success rate for a given tolerance of 10% pixel error (Fig. 4). We experimented with different contrastive approaches (triplet loss and multi-class n-pair loss [48]) and InfoNCE empirically performed the best in terms of stability and suitability of the representations for registration. Therefore we focused on InfoNCE for this study.

**Reviewer 4:** We compared our approach with approaches that do not require paired images (MI and GANs) and showed that they were outperformed by our method. In cases where aligned pairs cannot be obtained by acquisition, manual registration or computational means, non learning-based approaches such as MI remain the best option available.

[Meta-Review · NeurIPS 2020]

Overall, this paper received mixed reviews. The main argument against this paper (raised by both reviewer R1 and R3) is its novelty, as it currently seems to be primarily a variant of the infoNCE objective using data-augmentation that capture rotational invariances. After having read the reviews, author response and followed the reviewer discussion I would like to recommend acceptance. The main reason for this is that, as R 1 and R3 mention, this is a well-executed paper (perhaps better seen as an application paper), with good experiments and evaluation and also a method that tested and worked in 2 datasets, one of which in the biomedical domain.